

# A novel approach to improving colonoscopy learning efficiency through a colonoscope roaming system: randomized controlled trial

Dandan Ning[1], Huiyong Geng[2], Jingming Guan[1], Sijia Zhang[3], Shuang Wang[1], Shuang Li[1] and Shizhu Jin[1]

[1] Gastroenterology and Hepatology, The Second Affiliated Hospital of Harbin Medical University, Harbin Medical University, Harbin, Heilongjiang Province, China
[2] Animation Faculty, Harbin University of Science and Technology, Harbin, Heilongjiang Province, China
[3] Education Office, The Second Affiliated Hospital of Harbin Medical University, Harbin Medical University, Harbin, Heilongjiang Province, China

Corresponding author
Shizhu Jin, drshizhujin@hrbmu.edu.cn

## ABSTRACT

**Background**. Colonoscopy is indispensable in the diagnosis and treatment of lower digestive tract (LDT) diseases. Skilled colonoscopists are in great demand, but it takes considerable time for beginners to become experts. In addition, patients may refuse to permit primary learners to practise colonoscopy on them. Thus, improving the instructional programmes and models for primary learners is a key issue in endoscopy training. Convenience and a self-paced, learner-centred approach make e-learning an excellent instructional prospect. Therefore, we created the Colonoscope Roaming System (CRS) to assist in colonoscopy teaching procedures. We aimed to develop the e-learning software, test it with beginner colonoscopists and evaluate its effectiveness via subjective and objective methods.

**Methods**. Through a randomized controlled trial, participants were randomly allocated to an e-learning group (EG) or a control group (CG) after a pretest evaluation. The CG learned through the traditional colonoscopy teaching mode, while the EG used CRS in addition to the traditional teaching mode. Subsequent to the training, the participants completed a posttest and colonoscopy examination. The EG also completed a satisfaction questionnaire.

**Results**. Of the 84 participants, 81 (96%) finished the colonoscopy learning and evaluation modules of the CRS. No conspicuous differences in the pretest scores were found between the EG and CG ($p > 0.05$). Two months later, the posttest scores for the EG were higher than those of the CG ($p < 0.001$), and the EG had better performance on the colonoscopy examination ($p < 0.01$). Overall, 86.25% of questions raised in Q1-Q20 were satisfied with the CRS and considered it successful.

**Conclusions**. The use of CRS may be an effective approach to educate beginner colonoscopists to attain skills.

## BACKGROUND

Colonoscopy is a conventional and indispensable method for the diagnosis and treatment of lower digestive tract (LDT) diseases. While the global demand for experts is enormous, the endoscopy training process for beginners to gain mastery as specialists is lengthy (*Li et al., 2020*). This training was further complicated by the spread of coronavirus disease 2019 (COVID- 19), which disrupted traditional work, life and learning (*Okereke et al., 2020*; *Sawarkar, Sawarkar & Kuchewar, 2020*; *Shrestha et al., 2020*; *Bączek et al., 2021*; *Baral & Baral, 2021*; *Elsalem et al., 2021*; *Singh et al., 2021*). Traditionally, experienced endoscopists have played an important role in training primary learners, while the primary learners have worked with their tutors to practice endoscopy on patients. It is common for primary learners to feel nervous and anxious and to be unable to obtain satisfactory cooperation from their patients. Traditional clinical training may fail to provide adequate information to meet learners' needs and compensate for their cognitive deficits. Thus, improving the instructional programmes and models for primary learners is key for successful endoscopy training.

Many researchers are studying e-learning applications in the field of medical education (*Sait & Tombs, 2021*; *Soundy et al., 2021*; *Su et al., 2021*; *Vedi & Dulloo, 2021*; *Oropesa et al., 2022*). The colonoscopy teaching model will inevitably develop in this direction due to digitization, networking and intelligence (*Feng et al., 2013*; *Longhini et al., 2021*; *Ouajdouni, Chafik & Boubker, 2021*). Thus, we created the Colonoscope Roaming System (CRS), an e-learning system designed to assist in colonoscopy training.

Our objectives were to (1) develop a colonoscopy e-learning system for student learning, (2) evaluate the system's effectiveness, and (3) promote the use of the software in medical education.

## MATERIALS & METHODS

### Recruitment

We recruited participants in hospitals by posting notices about the experiment.

Inclusion criteria were as follows: (1) voluntarily participate in the study and sign the informed document; (2) be able to complete gastroscopy independently without systematic study of colonoscopy; (3) own computer that can support CRS operation and have basic computer operation skills; and (4) complete the pretest questionnaire.

Exclusion criteria (those who met one of the criteria were excluded): (1) quit during the experiment or fail to complete the corresponding tasks as required; (2) the score of the pretest was too high (the accuracy rate was higher than 80%) because the CRS was designed for beginners who had not previously mastered colonoscopy knowledge (*Nakanishi et al., 2017*).

The experiment was carried out from May 2020 to May 2021, with Fig. 1 illustrating the flow. Following the 30-minute pretest by the 84 eligible volunteers, elimination was performed based on the exclusion criteria. IBM SPSS version 24 was employed to facilitate the random assignment of the qualified participants into two groups: the e-learning group (EG) and the control group (CG: non-e-learning). At the on-campus Endoscopy Centre

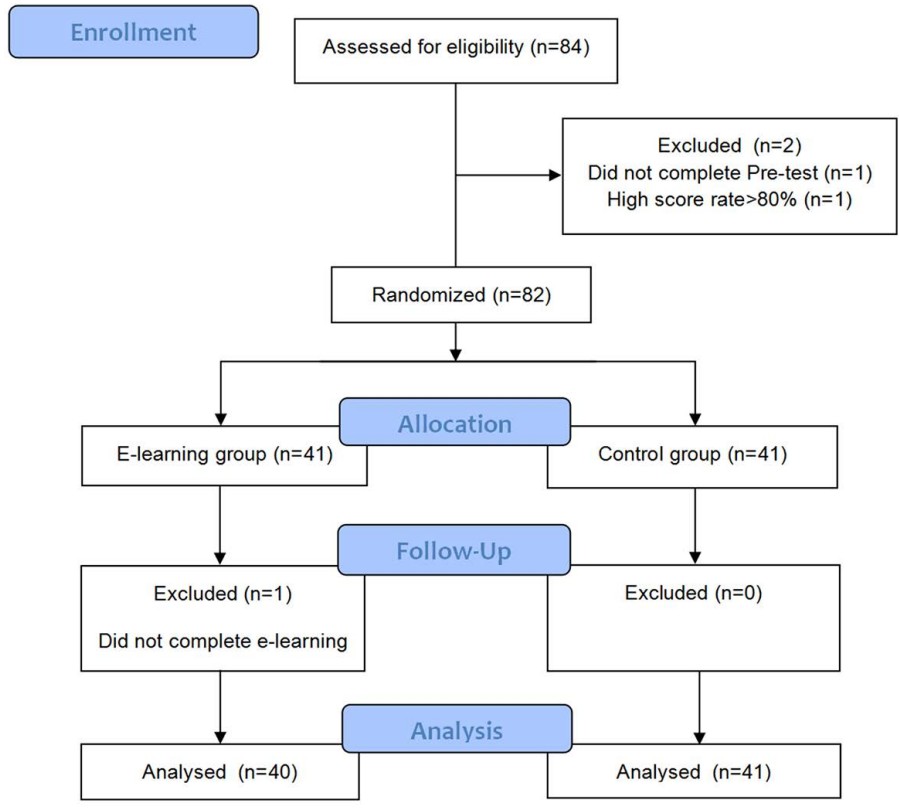

**Figure 1** Flow diagram showing the details of participant enrollment, randomization, and tests.

of the Second Affiliated Hospital of HMU, all enrolled students completed 2 months of theoretical and practical training in colonoscopy. In the process, each EG participant received an account number and corresponding password to facilitate logging into the CRS during the trial. Following the training, the study subjects completed the posttest and colonoscopy examination assessment two months later. The EG participants also completed a satisfaction questionnaire regarding CRS. This study was approved by Harbin Medical University Institutional Research Board (Approval Number HMUIRB2019008PRE). Participants gave written informed consent to participate in the study.

## Developing CRS and its description

We developed CRS in collaboration with Harbin University of Science and Technology. The specific development process is shown in Fig. 2. First, a 3-dimensional model (Figs. 3A–3B) of the LDT was obtained after a review of various data sources (mostly photographs and films of colonoscopy examinations from the Digestive Endoscopy Centre of the Second Affiliated Hospital of Harbin Medical University (HMU)). The initial structure of the model was created using Autodesk Maya software (Figs. 3C–3H), while the detailed structure of the colon cavity was finalized with ZBrush software (Fig. 3I). The 3D model was then transformed into an operating scene to simulate and render the physiological functions of the LDT. Second, the user interface of the system, including the log-in interface (Fig. 4A),

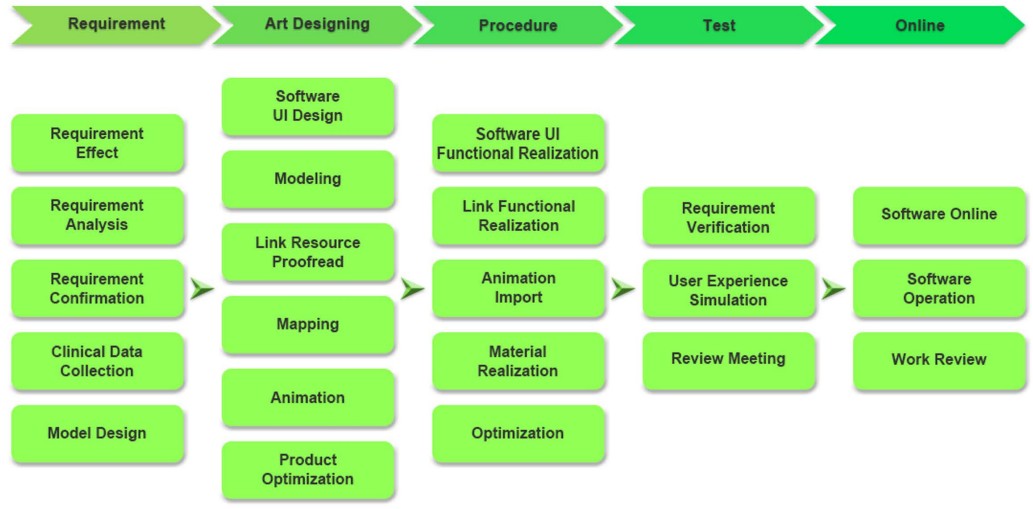

**Figure 2  The software design process.**

scene interface (Fig. 4B), exit interface, *etc.*, was designed using Unity 3D software. The C# programming language was used, giving the CRS the functions of interacting, cognition, and roaming (Figs. 4C–4G). CRS was built to be bilingual in Chinese and English and consists of two modes: focal trigger mode (mode A) (Fig. 4H) and structural exploration mode (mode B) (Fig. 4I).

## Learning outcome evaluation strategy

In the 1950s, Kirkpatrick designed a well-known framework for the evaluation of learning effects (*Giray, 2021*) that could be used to assess the effectiveness of the CRS (*Alliger & Janak, 1989*; *Ward et al., 2001*; *Galloway, 2005*). The framework encompasses these four aspects of assessment: reaction, learning, behaviour, and results (*Abdelhai et al., 2012*).

## Reaction level scoring

The reaction level was assessed through a satisfaction questionnaire. We used a questionnaire developed by *Wang (2003)*, which has been widely cited in evaluations of e-learning satisfaction. This questionnaire has been shown to have a reliability (Cronbach's alpha) of 95. Since the current CRS does not contain the function of online mutual communication, the evaluation section of the 'learning community' in the original questionnaire was removed in this experiment. The 20 questions were divided into four modules: content satisfaction, interface satisfaction, test satisfaction and personalized learning satisfaction. A 5-point Likert scale was applied to score each question (strongly disagree 1, disagree 2, neutral 3, agree 4, strongly agree 5). The students' evaluation of each module and the overall evaluation were calculated. All EG participants finished the satisfaction questionnaire after training completion. Because of the COVID-19 pandemic and the paperless procedures, we created quick response (QR) codes for the participants to answer the questions. The QR codes and the questionnaire are provided in File S1.

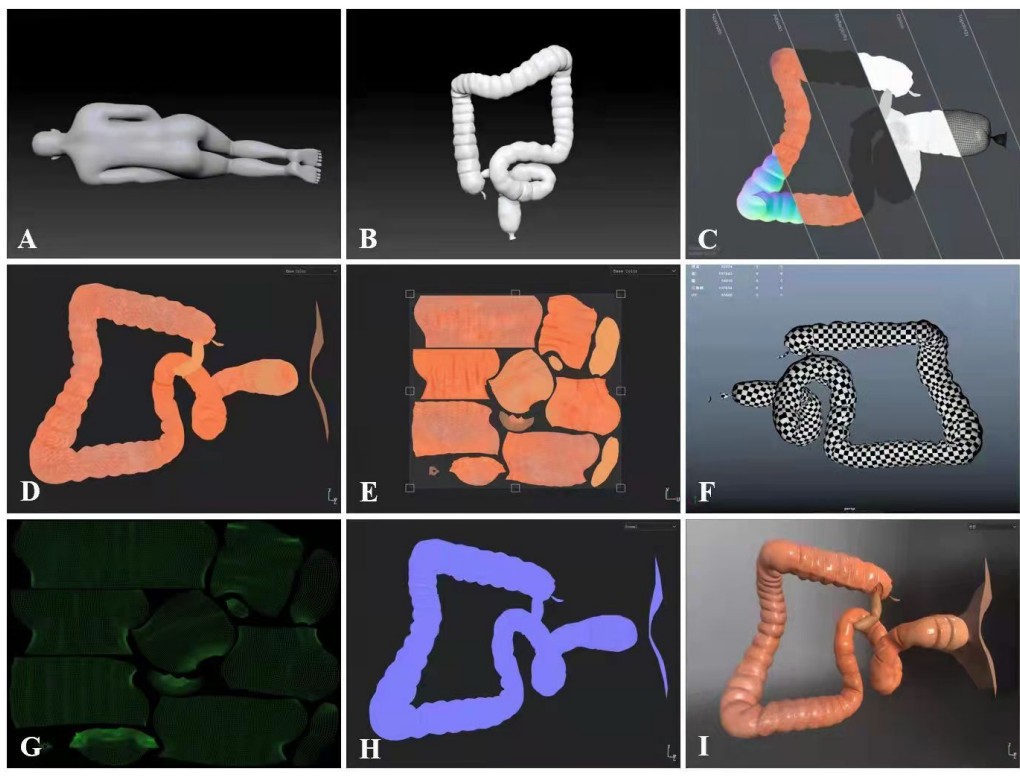

**Figure 3** **CRS software development of screen captures.** (A–B) 3-dimensional model. (C–H) Bake mapping technique from left to right for normal mapping, ambient occlusion, reflective mapping, gloss mapping and topology. (I) 3D model screenshot.

## Assessing the learning level

The most routine and reasonable way to assess learning levels is to compare test scores before and after the intervention. As mentioned above, the pretest was performed prior to colonoscopy learning, and the posttest was performed following the 2 months of training. To ensure that the pretest and posttest are of the same difficulty, both tests have the same questions, but the order of the questions is different (*Oliveira, Mattos & Coimbra, 2017*). All of the colonoscopy images used in the questions were from the Department of Gastroenterology, the Second Affiliated Hospital of HMU. The test was compiled by experts from our endoscopy centre. The QR codes and the test (posttest version) are available in File S2.

## Behaviour level

As routine colonoscopy training was provided to all participants, we assessed behavioural levels by measuring participants' actual colonoscopy performance. All the patients participating in the colonoscopy tests had made appointments for a general colonoscopy at the outpatient department and had agreed to have their colonoscopies performed by beginner colonoscopists. All students needed to perform 20 colonoscopies after completing the entire learning process. During the procedures, the instructor recorded the student's

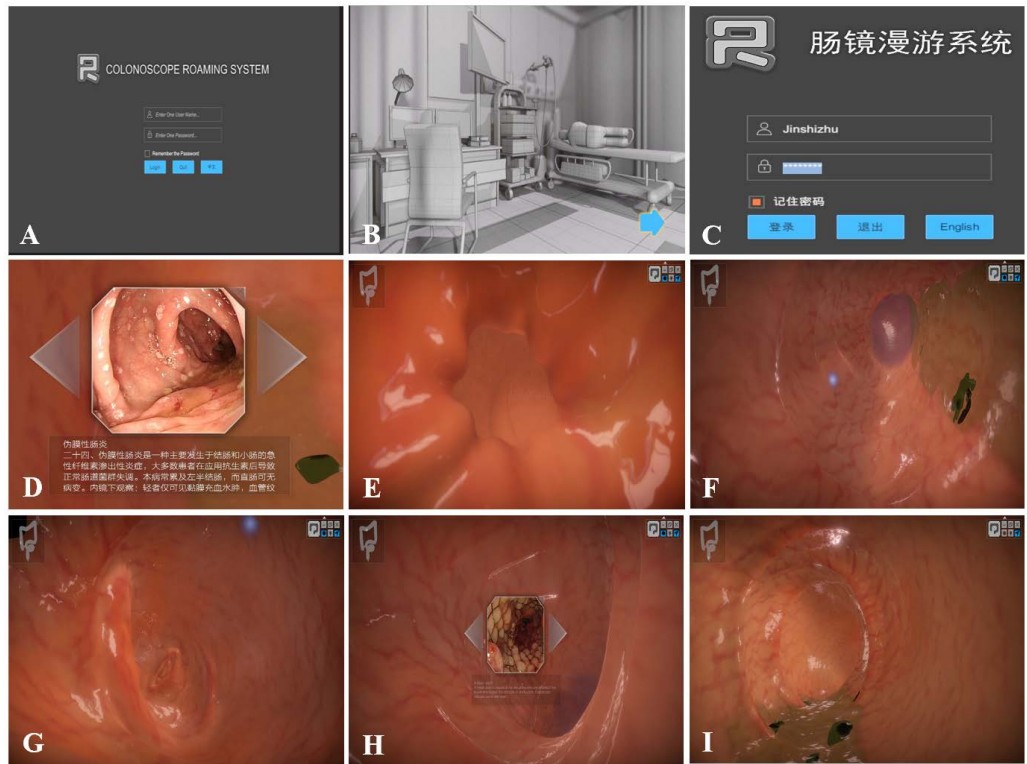

**Figure 4** **CRS software screen captures.** (A) Software system log-in interface in English. (B) Scene interface. (C) Software system log-in interface in Chinese. (D–I) Initial screen of the software.

procedures, and the video was used to assess the following: (1) the times to pass through the LDT and to reach the sigmoid colon, splenic flexure, hepatic flexure and cecum; (2) the overall colonoscopy examination time and fluency; (3) the clarity of the collected images; (4) diagnostic accuracy; (5) the patient's comfort and satisfaction; and (6) the detection rate of colonic polyps. After three failed attempts in the above procedures, the expert endoscopists performed the procedures and recorded it as a failure for the student (*Cass et al., 1993*). The entire operation process of the trainees was video-recorded. Each participant had an opportunity to select the performance that he or she thought was the most satisfactory for archiving. A total of five expert colonoscopists (from the Second Affiliated Hospital of HMU) who were unaware of the participants' grouping scored the students' colonoscopy examination videos based on their experience in colonoscopy. The experts collaborated to develop the examination standards. The mean score was obtained. The QR codes and the grading table for colonoscopy examinations are provided in File S3.

### Result level scoring

To evaluate the results, we developed a course called 'Integrated Online and Offline Teaching of Digestive Endoscopy Class' for the standardized training of residents at the Second Affiliated Hospital of HMU. The course certificate is shown in File S4.

## Statistical analysis

R Studio 3.6.3 was used for data analysis. Measurement data following a normal distribution are described as the means and standard deviations; measurement data not following a normal distribution are described as the medians and upper and lower quartiles. For measurement data following a normal distribution that had homogeneous variance between groups, Student's t test was employed for comparisons between the two groups. If data did not follow a normal distribution or the variances between the two groups were not uniform, the comparison between the groups was performed by using the nonparametric Wilcoxon rank sum test. Enumeration data are described as the frequencies (%), and the $\chi 2$ test was used for intergroup comparisons. A $P$ value of $< 0.05$ was considered to be statistically significant.

## RESULTS

### Sample description

Of the 84 eligible students from the Second Affiliated Hospital of HMU, two participants were excluded based on the exclusion criteria. One student secured $> 90\%$ in the pretest, and another did not complete the pretest. Eighty-two (97%) fit the inclusion criteria after completing the pretest. Random assignment of all 82 participants was done to the EG and the CG on average. In the EG, one dropped out during the training period. Therefore, there were 40 participants in the EG and 41 participants in the CG. Both groups displayed similar baseline characteristics, as depicted in Table 1.

### Analysing the examination scores

The pretest and posttest scores in the EG group and the CG group were not normally distributed, so they are represented as the medians and upper and lower quantiles. A nonparametric Wilcoxon rank sum test was used to perform comparisons between the two groups. The pretest score in the EG was 48 (48, 50) and increased to 83 (80, 86) on the posttest ($p < 0.0001$). In contrast, the pretest score in the CG was 50 (46, 50) and increased to 80 (76, 84) on the posttest ($p < 0.0001$). The results showed that there were no significant differences in the pretest scores between the EG and CG ($p > 0.05$). In addition, after 2 months of training, the posttest results of the EG were better than those of the CG ($p < 0.001$) (Fig. 5A).

For the colonoscopy examination scores, a total of 800 colonoscopies were performed among 40 students in the EG. There were 617, 559, 526 and 275 instances of successful passage of the colonoscope through the sigmoid colon, splenic flexure, hepatic flexure and cecum, respectively. In the CG, 41 students performed a total of 820 colonoscopies, among which there were 587, 526, 491 and 229 successful instances of passage of the colonoscope through the sigmoid colon, splenic flexure, hepatic flexure and cecum, respectively. There were statistically significant differences in the number of successful instances of passage of the colonoscope through the sigmoid colon, splenic flexure, hepatic flexure and cecum between the EG and the CG, with the EG having a higher number of successful instances than the CG (Fig. 5C). The detection rates of colon polyps in the EG and CG were 66 (8.25%) and 38 (4.63%), respectively, which was a significant difference ($p < 0.01$) (Fig.

**Table 1  Demographics and other characteristics of the participants.**

| Baseline characteristics | E-learning group (n = 40) | Control group (n = 41) | $X^2$/W | p value |
|---|---|---|---|---|
| Age (years) | 25 (25, 25.25) | 25 (25, 26) | 751 | 0.4263 |
| Gender, n (%) | | | | |
|    Male | 14 (35%) | 15 (36.59%) | 0.0221 | 0.8817 |
|    Female | 26 (65%) | 26 (63.41%) | | |
| Current year of residency training, n (%) | | | | |
|    Postgraduate year 1 | 6 (15%) | 5 (12.20%) | | |
|    Postgraduate year 2 | 22 (55%) | 24 (58.54%) | 0.1656 | 0.9206 |
|    Postgraduate year 3 | 12 (30%) | 12 (29.26%) | | |

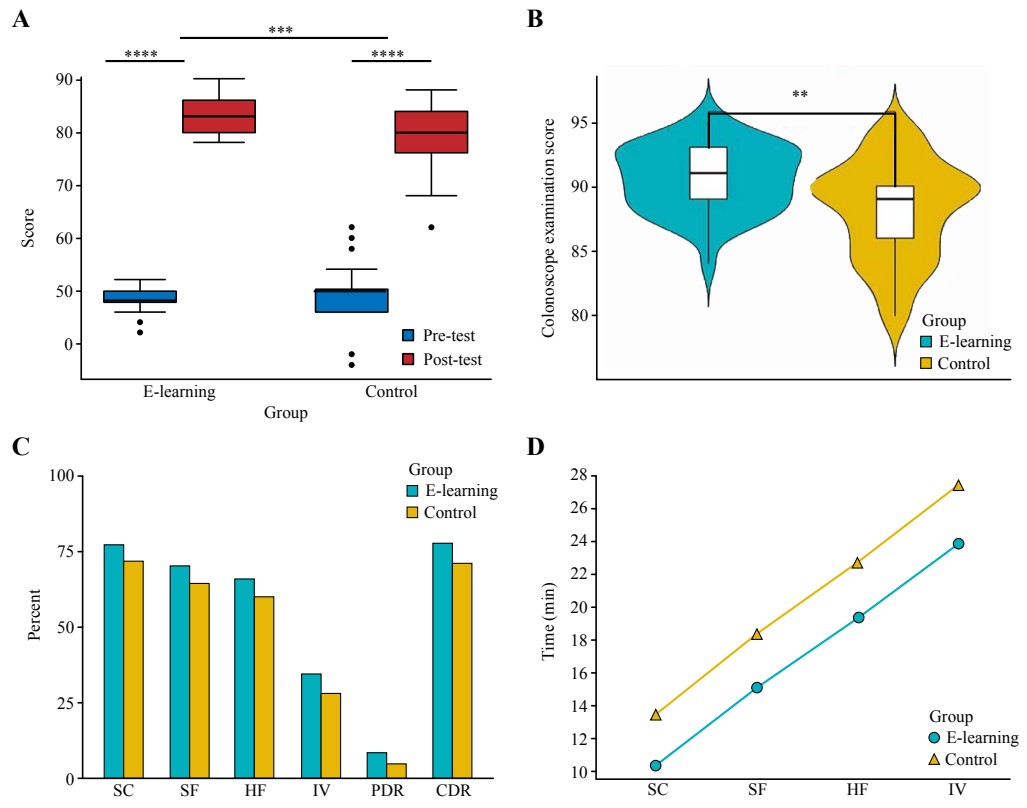

**Figure 5  Learning and behavior level assessing.** (A) Pretest and posttest scores of the e-learning group and control group. (B) Colonoscope examination results of the e-learning group and control group. (C) The success rate of passage of the colonoscope through the sigmoid colon (SC), splenic flexure (SF), hepatic flexure (HF) and cecum (IV); polyp detection rate (PDR); and correct diagnosis rate (CDR). (D) The time period to reach each segment of the intestinal cavity in the EG and CG. ** $p < 0.01$, *** $p < 0.001$, **** $p < 0.0001$.

5C). The detection rate of polyps in the EG was higher than that in the CG. The numbers of correct diagnoses during colonoscopy in the EG and CG were 620 (77.5%) and 581 (70.85%), respectively, which was a statistically significant difference ($p < 0.01$) (Fig. 5C).

**Table 2  Examination scores.**

| | E-learning group | Control group | $X^2/t$ | $p$ value |
|---|---|---|---|---|
| Sigmoid colon, n (%) | | | | |
|     Successful | 617 (77.12%) | 587 (71.59%) | 6.5112 | 0.0107 |
|     Unsuccessful | 183 (22.88%) | 233 (28.41%) | | |
| Splenic flexure, n (%) | | | | |
|     Successful | 559 (69.87%) | 526 (64.15%) | 6.0082 | 0.0142 |
|     Unsuccessful | 241 (30.13%) | 294 (35.85%) | | |
| Hepatic flexure, n (%) | | | | |
|     Successful | 526 (65.75%) | 491 (59.88%) | 5.9751 | 0.0145 |
|     Unsuccessful | 274 (34.25%) | 329 (40.12%) | | |
| Cecum, n (%) | | | | |
|     Successful | 275 (34.38%) | 229 (27.93%) | 7.8559 | 0.0051 |
|     Unsuccessful | 525 (65.62%) | 591 (72.07%) | | |
| Detection rate of polyps, n (%) | | | | |
|     Cases of polyps | 66 (8.25%) | 38 (4.63%) | 8.8127 | 0.0030 |
|     None | 734 (91.75%) | 782 (95.37%) | | |
| Instances of correct diagnosis, n (%) | | | | |
|     Correct | 620 (77.50%) | 581 (70.85%) | 9.3288 | 0.0023 |
|     Incorrect | 180 (22.50%) | 239 (29.15%) | | |
| Colonoscopy training score | 91 (89, 93) | 89 (86, 90) | 1130 | 0.0033 |

The correct rate of disease diagnosis in the EG was higher than that in the CG (Fig. 5C). The colonoscopy training score of the students in the EG was 91 (89, 93) points and that of the CG was 89 (86, 90) points. The difference between the two groups was statistically significant, with the EG having a higher score than the CG (Fig. 5B). The time to reach the sigmoid colon, splenic flexure, hepatic flexure and cecum in the EG and the CG were determined, and the time to reach each segment of the colon was shorter in the EG (Fig. 5D). Table 2 summarizes the examination scores.

## User satisfaction of the CRS

The satisfaction levels are detailed in four areas. Regarding the content quality, 87% of questions raised in Q1-Q5 demonstrated that the course was up-to-date, useful, and sufficient and fit their needs (Fig. 6A). Regarding the interface quality, 83% of questions raised in Q6-Q10 showed that the students were in favour of the speed, stability and user-friendly approach with ease of finding the necessary content (Fig. 6B). Regarding the testing quality, 86.5% of questions raised in Q11-Q15 reveled that the course testing was prompt, secure, fair and easy to complete and understand (Fig. 6C). Regarding the quality of personalization, 88.5% of questions raised in Q16-Q20 claimed that the CRS facilitated personalized learning, controlled their learning progress, selected what to learn, recorded their learning progress, and helped them learn the necessary content (Fig. 6D). Overall, 86.25% of questions raised in Q1-Q20 were satisfied with the CRS and considered it successful.

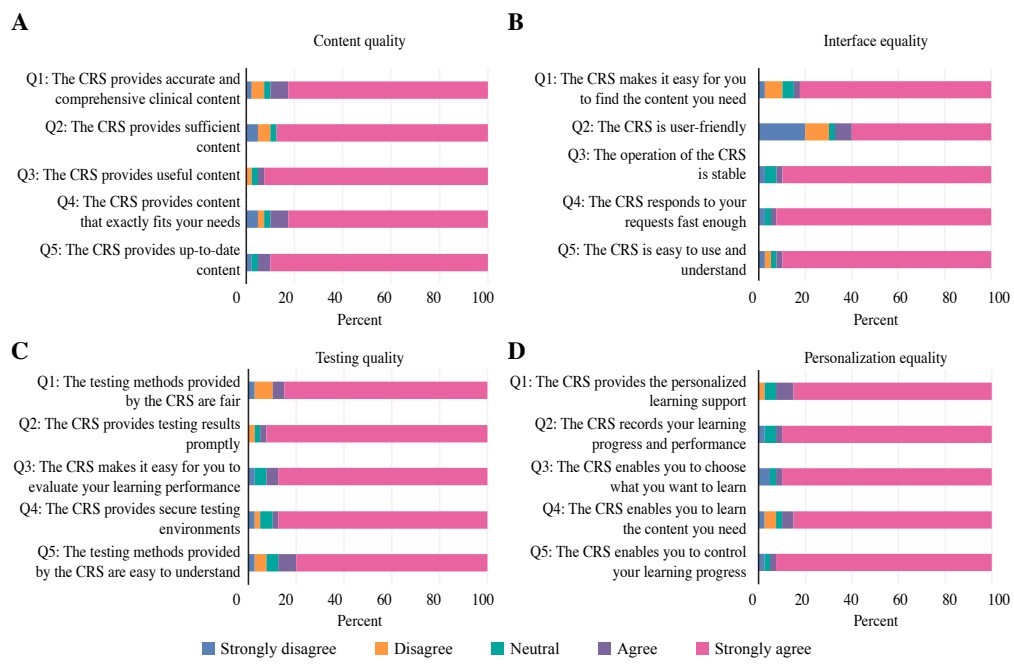

**Figure 6** **Analysis of the participants' satisfaction with the CRS in four areas.** (A) Content quality. (B) Interface quality. (C) Testing quality. (D) Personalization quality.

**Table 3** **Demographics and other characteristics of the patients.**

| Baseline characteristics | E-learning group ($n = 800$) | Control group ($n = 820$) | $X^2$ | p value |
|---|---|---|---|---|
| Age (years) | 53(50,57) | 53(50,56) | 47.980 | 0.847 |
| Gender, n (%) | | | | |
| Male | 411 (51.38%) | 427 (52.07%) | 0.0790 | 0.7786 |
| Female | 389 (48.62%) | 393 (47.93%) | | |
| History of abdominal operation | | | | |
| Yes | 357(44.63%) | 368(44.88%) | 0.0105 | 0.9184 |
| No | 443(55.37%) | 452(55.12%) | | |

To further validate students' feedback, we conducted a satisfaction survey with patients. Both groups displayed similar baseline characteristics, as depicted in Table 3. All the patients in the e-learning group participated in the satisfaction survey, which showed a high level of satisfaction (Fig. 7).

# DISCUSSION

## Principal results

To date, textbooks and other written materials are considered the most widely spread knowledge in traditional colonoscopy training (*Trelease, 2016*), but in the field of medical skill operations, the traditional mode does not provide three-dimensional thinking of the

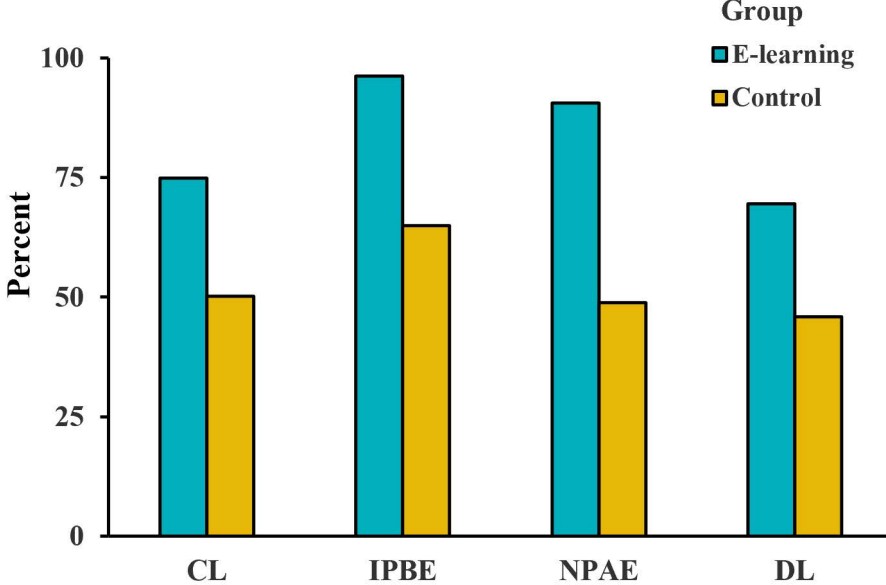

**Figure 7** **Analysis of the patient satisfaction questionnaire with the CRS in four areas.** (A) Comfort level (CL). (B) Inform the process before examination (IPBE). (C) Notice of precautions after examination (NPAE). (D) Diagnostic level (DL).

digestive tract structure and operational experience. In addition, the COVID-19 pandemic has necessitated mandatory e-learning in medical education. In this study, we developed, evaluated and applied e-learning software named the CRS during the COVID-19 pandemic. We developed the CRS for student learning colonoscopy in collaboration with Harbin University of Science and Technology, with the CRS being the first e-learning program for colonoscopy teaching. Furthermore, the CRS was developed to be used in either Chinese or English; therefore, the system can facilitate colonoscopy training in China and worldwide.

After two months of learning, the theoretical test scores of all the students significantly increased, with an EG score greater than that of the CG ($p < 0.001$) (Fig. 5A). This shows that CRS is more effective in improving theoretical knowledge. The success rate of EG was higher than that of CG in the four key points of colonoscopy operation, namely, the sigmoid colon, splenic curvature, hepatic flexure and cecum. CRS provides beginners with a highly simulated virtual environment of LDT, enabling them to freely explore the structure of LDT and then fully master it through repeated practice so that they can have a good understanding of the actual operation and effectively improve the success rate of operation. In the endoscopic diagnosis and the polyp detection rate, the EG was significantly higher than the CG ($p < 0.01$), indicating that the students had enriched their experience in disease identification by learning endoscopic images of common LDT diseases embedded in CRS. We further submitted the operation videos of the students to expert colonoscopists for scoring and found that the scores of EG were higher than those of CG ($p < 0.01$), which fully demonstrated that the success rate of colonoscopy operation can be improved through CRS in a short period of time. It was verified that CRS enabled beginners to learn

colonoscopy better and faster after they fully mastered the anatomical structure of LDT, theoretical knowledge and endoscopic manifestations of common diseases.

However, the CRS is not meant to be a complete colonoscopy training but a complementary training. E-learning is not a substitute for traditional learning, but an effective supplement to be used in combination with traditional learning mode.

The results of the satisfaction questionnaire show that the overall satisfaction rate of the students in the EG was very high (Fig. 6). Based on Wang's questionnaire, we obtained satisfaction information with each aspect of the training. In terms of interface quality (Fig. 4B), the CRS's operating interface was considered complex and somewhat difficult to understand. The reason may be that the software design uses 'Z/X' to represent inhalation and inflation, respectively. In the process of the insertion and passage of the colonoscope through the digestive tract, the intestinal cavity must be continuously inflated; otherwise, the colonoscope cannot pass through. We upgraded and improved the software version based on feedback from the trainees and added detailed instructions for the procedures.

### Limitations

First, the small sample size at a single institution affects the generalizability of the results. The main reason for the current study design was that our CRS software was still in its initial stage. In the future, we will integrate endoscopy examination and treatment to establish a complete endoscopy e-learning teaching system. Residents should be promoted to this system through standardized training. Second, during the evaluation of the actual colonoscopy procedures, there was no way to ensure the same difficulty level among students because of the variations in the LDT structure and the differences in pain tolerance in each patient.

## CONCLUSIONS

This experiment shows that CRS, as a meaningful supplementary approach to colonoscopy training, is valid. The e-learning program can help beginners master colonoscopy skills and become familiar with the characteristics of LDT. User satisfaction with CRS was extremely high among the participants in this study. In summary, the use of the CRS may be an effective method to educate beginner colonoscopists to attain skills and recognize the symptoms of common LDT diseases.

## ACKNOWLEDGEMENTS

The authors thank AJE for editing this manuscript.

### Funding

This work was supported by the Heilongjiang Province Education Science Planning Key Project under Grant (No. GJB1320190) and the Harbin City Eagles Project under Grant (No. 2019CYJBCG0339). The funders had no role in study design, data collection and analysis, decision to publish, or preparation of the manuscript.

## Grant Disclosures

The following grant information was disclosed by the authors:

The Heilongjiang Province Education Science Planning Key Project: GJB1320190.

The Harbin City Eagles Project under Grant: 2019CYJBCG0339.

## Competing Interests

The authors declare there are no competing interests.

## Author Contributions

- Dandan Ning performed the experiments, prepared figures and/or tables, contributed analysis tools, and approved the final draft.
- Huiyong Geng performed the experiments, performed the computation work, authored or reviewed drafts of the article, and approved the final draft.
- Jingming Guan analyzed the data, prepared figures and/or tables, and approved the final draft.
- Sijia Zhang analyzed the data, prepared figures and/or tables, and approved the final draft.
- Shuang Wang performed the experiments, prepared figures and/or tables, authored or reviewed drafts of the article, and approved the final draft.
- Shuang Li performed the computation work, prepared figures and/or tables, and approved the final draft.
- Shizhu Jin conceived and designed the experiments, authored or reviewed drafts of the article, and approved the final draft.

## Ethics

The following information was supplied relating to ethical approvals (i.e., approving body and any reference numbers):

The Harbin Medical University Institutional Research Board approval to carry out the study within its facilities (Ethical Application Ref:HMUIRB2019008PRE).

## Data Availability

The raw measurements are available in the Supplemental Files and images are available at figshare: Ning, DanDan (2022): data. figshare. Dataset. https://doi.org/10.6084/m9.figshare.18866717.v3.

## Supplemental Information

Supplemental information for this article can be found online at http://dx.doi.org/10.7717/peerj-cs.1409#supplemental-information.

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
