# Peer review of "A novel approach to improving colonoscopy learning efficiency through a colonoscope roaming system: randomized controlled trial"

_PeerJ Computer Science, doi:10.7717/peerj-cs.1409_

## Round 0.1 · original submission · Major Revisions

It has been a tough process due to very limited pool of experts who could be reached out to. A number of such reviewers also declined due to other prior commitments. We certainly appreciate authors' patience in this regard. I urge you to do a careful read of the reviewers' comments and incorporate the changes as suggested. Also, it will be great if you are able to separately submit a response addressing those comments.

·

Basic reporting

The writing is clear and easy to follow. However, there is no proper justification or formatting done. Not much literature has been mentioned using e-resources (CRS) for colonoscopy teaching. Is the motivation behind the work only justifiable during COVID? Can't it be used in normal scenario? If yes, then it should be clearly stated in the paper. No Hypothesis has been used to further justify the results. Correct the typos.

Experimental design

The research work done is satisfactory and would have been more meaningful had it been tested over 1000 samples. A good questionaire would further be meaningful in justifying the results. The analytical methods used for justifying the results are good.

Validity of the findings

The findings would have been more justifiable had been tested on more number of subjects. A good hypothesis with comparison with the existing schemes would have justified the accuracy of the results.

Reviewer 2 ·

Basic reporting

The authors developed an e-learning software designated for colonoscope roaming systems to assist in colonoscopy technique procedure. The aim of the study is to develop CRS and apply it with beginner colonoscopists for evaluating subjective and objective methods.

Experimental design

The study is a randomized trial where participants were randomly allocated to an e-learning group after a pretest evaluation.

Validity of the findings

Their study suggested a positive outcome of 86.25%. 690 out of 800 patients were satisfied with CRS. During the pandemic, it can be an alternative approach for e-learning and educating beginner colonoscopists to boost their skill.

Here are my comments:

1. The study is well-designed. However, from the reader's perspective, we would like to see more illustrations of the figures which is not present at the current stage.

2. Some more statistical validation is required. Currently, the author uses student t-test. There are a different tests for validating the approaches.

3. The data can be represented in a better way.

4. The authors are encouraged to put their information in Tables.

5. clinical motivation can be improved.

---

## Round 0.2 · Minor Revisions

This is an extract from the section on Results.

"Of the 84 participants, 81 (96%) finished the colonoscopy learning and evaluation modules of the CRS. No conspicuous differences in the pretest scores were found between the EG and CG (p>0.05). Two months later, the post-test scores for the EG were higher than those of the CG (p<0.001), and the EG had better performance on the colonoscopy examination (p<0.01). Overall, 86.25% (690/800) of the participants were satisfied with the CRS."

In my assessment, a better explanation & a firmer basis needs to be provided for the statistics cited. It has been pointed out by a reviewer that the 'sample size is small'. I would like to ascertain that and include a response as multiple times 'small sample size' tends to limit the validity of the research findings.

Your revised version ought to include explicitly the count for those who learned via ER & those who learnt via CG in this section. Same way, 800 participants are not the same as 84 participants stated in the first sentence. Please explain the basis for the terms such as 'scores for the EG were HIGHER...' and 'BETTER performance ...'. There is perhaps a matric for measuring these in a statistically significant/relevant manner. All the 'p' values must also be explained with clear reasoning.

On another point, you have only included gender/age information in demographics in Tables 1 & 3. Please clarify if any other demographic information (food habits, economic well-being, smoking/drinking habits, ethnicity for example) may have influenced the hypotheses being tested.

I thank you for your patience and look forward to the revised version.

·

Basic reporting

The authors have replied to the reviewer's comments and have successfully implemented them. The paper can now be accepted in its present form.

Experimental design

No comments

Validity of the findings

The work is novel however, the sample size is small.

Additional comments

No comments.

---

## Round 0.3 · Minor Revisions

It is truly the smallest revision that may be asked for. I think a more appropriate and hence acceptable conclusion is that "The use of CRS may be..." given all the complexities associated with the entire testing process, statistical nature of the work and the sample size. I would also think that if the authors agree and revise the manuscript accordingly, the manuscript may be accepted. Also, an apology is due for the extended time it has taken to carry out the review.

---

## Round 0.4 · accepted · Accept

It has taken us several rounds of reviews. However, we all were fully engaged into ensuring that the work is well articulated and comprehensive. Congratulations!